# Suicidal Ideation in Adolescents and Young Adults: The Role of Defeat, Entrapment, and Depressive Symptoms—From a Systematic Review to a Tentative Theoretical Model

**DOI:** 10.3390/bs14121145

**Published:** 2024-11-28

**Authors:** Ana Huertes-del Arco, Eva Izquierdo-Sotorrío, Miguel A. Carrasco, Irene Caro-Cañizares, Francisco Pablo Holgado-Tello

**Affiliations:** 1Department of Personality, Assessment and Psychological Treatments, Faculty of Psychology, National University of Distance Educationa (UNED), 28040 Madrid, Spain; ahuertes3@alumno.uned.es (A.H.-d.A.); pfholgado@psi.uned.es (F.P.H.-T.); 2School of Health Sciences and Education, Open University of Madrid (UDIMA), 28400 Madrid, Spain; eva.izquierdo@udima.es (E.I.-S.); irene.caro@udima.es (I.C.-C.)

**Keywords:** entrapment, defeat, depressive symptoms, suicidal ideation, adolescents, young adults, systematic review

## Abstract

Suicide stands as one of the leading causes of non-accidental death among adolescents and young adults. The Integrated Motivational–Volitional Model (IMV) of suicidal behavior identifies feelings of defeat and entrapment as pivotal factors in the complex dynamics underlying suicidal behaviors. Additionally, depressive symptomatology plays a crucial role in the development of these behaviors. The aim of this work was to provide a systematic review of the scientific literature on the association of three risk factors—entrapment, defeat, and depressive symptomatology—with suicidal ideation in the adolescent and young adult population and to test a tentative theoretical model. The databases used were PubMed, Web of Science, and ProQuest. Primary studies were selected that examined the role of entrapment, defeat, and depressive symptomatology in suicidal ideation among adolescents and young adults (ages 10–35). Additionally, a model based on structural equation modeling was analyzed to assess the relationships between entrapment, defeat, and depressive symptomatology and suicidal ideation. Eleven studies met the established inclusion criteria. The results showed defeat and entrapment to be associated with depressive symptomatology and suicidal ideation, regardless of the country studied. Additionally, the model indicates that entrapment and defeat lead to depressive symptomatology, which in turn contributes to the occurrence of suicidal ideation. These findings suggest that defeat and entrapment could be universal factors in explaining suicidal behavior in adolescents and young adults. Consequently, the results of this work may contribute to the development of strategies for preventing suicidal ideation in this population.

## 1. Introduction

According to the World Health Organization [1], approximately 703,000 people die by suicide each year, making suicidal behavior one of the leading causes of death worldwide. It has become the second leading cause of premature death among young people aged 15 to 29 [2].

Suicidal ideation and behaviors typically begin to emerge during preadolescence and become more pronounced during adolescence [3,4,5]. It is estimated that adolescents experiencing suicidal ideation are 12 times more likely than others to attempt suicide by the age of 30 [3,4,5]. Furthermore, suicidal ideation and attempts in adolescents are associated with a lower life satisfaction and reduced emotional well-being [6].

Suicidal behavior is a complex and multifaceted reality that appears to be influenced by numerous biopsychosocial factors [7]. Additionally, the existential and social nature of suicide complicates its reduction into discrete elements [8]. For this reason, new approaches to the study of suicide propose a shift in focus, from searching for the causes of suicide to understanding the reasons and processes involved, with the aim of developing effective prevention strategies [9].

The Integrated Motivational–Volitional (IMV) model is one of the current theoretical paradigms seeking to explain suicidal behavior [10,11]. According to this model, suicidal behavior begins and develops through three phases: the **premotivational phase** (i.e., dispositional factors and triggering events), **the motivational phase** (i.e., suicidal ideation and intent), and the **volitional phase** (i.e., suicidal action). Throughout these phases, various factors are considered to be responsible for the onset of suicidal ideation and its progression towards attempted and completed suicide. The IMV model considers the perception of feeling defeated combined with a sense of entrapment and a lack of available help as key factors in the progression from passive ideation, or the emergence of thoughts of ending one’s life as a sole means of escape, to the transition to an attempt. Defeat has been defined [12] as feelings of helplessness and humiliation along with a perceived failure in a struggle following a loss of or change in a valuable status of the individual. Meanwhile, entrapment is characterized as the sensation of being unable to escape from painful or confining situations [13].

One of the main elements of interest in O’Connor’s model [10,11] is its proposal of the mechanisms that contribute to the transition from suicidal ideation to lethal suicidal action (motivational–volitional phase). Three groups of moderators are identified in this transition. In the motivational phase, moderators include deficits in problem-solving, rumination, and biases in autobiographical memory, which catalyze the progression from defeat to entrapment. In the volitional phase, factors such as perceived burdensomenessand the perception of being a burden moderate the relationship between entrapment and suicidal ideation, influencing either the intensification of entrapment or the generation of more hopeful expectations. Within the motivational phase, depressive symptomatology is highly relevant [14,15,16]. Notably, cognitive biases, negative attributions about the future, and hopelessness are prominent variables in the processes of suicidal ideation and crisis.

Previous studies have demonstrated a strong association between depressive symptomatology and suicidal behavior [17,18]. Other research has provided evidence of the association between hopelessness—a motivational factor in the IMV model [10] as well as a central concept in the work of Beck et al. [19,20]—and suicidal ideation [21,22]. A positive and statistically significant association has also been found between the severity of depressive symptoms, hopelessness, and suicidal ideation e.g., [12,23,24,25,26].

Meanwhile, Griffiths et al. [27] noted that depressive symptoms and defeat–entrapment variables can mutually reinforce each other and that both defeat–entrapment and depressive symptomatology are predicted in both directions. There is significant empirical evidence showing that both defeat and entrapment are predictors of suicidal ideation [28,29,30] and suicide attempts [31]. Additionally, the meta-analysis by Siddaway et al. [26] revealed a robust relationship between entrapment, defeat, and suicidal risk.

Given the prominent role of these variables in suicidal ideation in the general population [32,33,34,35,36,37,38], review studies that analyze different findings on their contribution to the prediction and prevention of suicidal behavior in community and clinical populations will be useful for understanding the current state of the issue and the mechanisms leading to suicidal ideation and behavior. Additionally, exploring models of the relationship between entrapment, defeat, and depressive symptomatology with respect to suicidal ideation in the adolescent–young adult population may advance our understanding of suicidal behavior and guide more effective therapeutic strategies for its prevention.

The primary objective of this work is to explore the findings from empirical research on the role of entrapment and defeat in suicidal behavior among adolescents and young adults. A secondary aim, based on the data extracted from the review, is to investigate the potential mediating effect of depressive symptomatology in the relationship between defeat, entrapment, and suicidal ideation in this population.

## 2. Method

The search for studies was carried out in accordance with PRISMA-P (Preferred Reporting Items for Systematic Review and Meta-Analysis Protocols) [39,40].

The review protocol was registered in March 2024 with the International Prospective Register of Systematic Reviews (PROSPERO, registration number CRD42024525551) to ensure transparency in the review procedures undertaken [41].

The AMSTAR 2 [42] tool was used to assess the quality of the systematic review conducted.

### 2.1. Search Strategy

In March 2024, a systematic search was conducted using the electronic databases PubMed, Web of Science (Core Collection), and ProQuest. No limits were imposed on the publication date of the studies. The search was performed by two of the authors (A.H. and E.I.), who independently assessed the eligibility of each article. After an initial screening based on this criterion, the full texts of the remaining articles were examined to evaluate their potential inclusion.

The search strategy was conducted using a combination of the following terms: (1) “Entrapment” and “Defeat” and “Suicid*” and “adolescen*”; (2) “Entrapment” and “Defeat” and “Suicid*” and “young adults”.

The search terms were grouped into four sets using the Boolean term AND. To select the primary original documents, the following search string was used across all fields: [(“entrapment” AND “defeat” AND “suicid*” AND “adolescen*”)]. Simultaneously, a second search string was used that modified the target population: [(“entrapment” AND “defeat” AND “suicid*” AND “young adults”)]. In ProQuest (Ann Arbor, MI, USA), without a restricted search, the results were unmanageable; therefore, the search was conducted using the same strings but selecting the “Abstract” field for each term.

The eligibility criteria for the studies analyzed were as follows. The inclusion criteria called for primary studies that (a) involved participants who were adolescents and young adults of both sexes, aged between 10 and 35 years, from both clinical and general populations; (b) investigated the relationship between defeat and/or entrapment variables and suicidal behavior; (c) used validated assessment instruments with evidence of reliability; and (d) were published in English. The exclusion criteria ruled out studies with samples of adults over 35 years old or mixed-age samples (including both young adults and adults); studies that omitted fundamental methodological information in the abstract, such as the number or ages of participants; and studies that did not provide differentiated data regarding the study variables, i.e., defeat or entrapment in relation to suicidal behavior. Review articles, trials, case studies, and qualitative studies were also excluded. Finally, articles written in languages other than English were excluded.

In longitudinal studies with multiple measurement points for the variables, only the measures from the first assessment point were considered in order to avoid overlap.

Figure 1 illustrates the sequence of the search and selection procedure for the studies. In total, 92% of the studies found were excluded after all titles and abstracts had been analyzed, leaving 11 studies that met the established eligibility criteria. It is important to note that the study by Tarsafi et al. [43] includes two samples, which were analyzed independently due to the presence of two distinct population groups (from Iran and the USA).

### 2.2. Data Extraction and Coding

In accordance with Lipsey and Wilson [44], a coding protocol and an Excel spreadsheet were designed to ensure both systematic data coding and replicability. The following categories were coded, beginning with bibliographic data (a–c), continuing with participant-related variables (d–e) and methodological variables (f), and ending with coding results (g) for (a) authorship; (b) publication year; (c) country where the study was conducted and where the sample was obtained; (d) number, mean and standard deviation of age, respective percentages of female and male participants; (e) sample type (adolescents vs. young adults, general vs. clinical); (f) instruments used to assess defeat, entrapment, and depressive symptomatology; and finally (g) correlations between pairs of any of these variables: depressive symptomatology, entrapment, defeat, and suicidal ideation.

### 2.3. Data Analysis

To be able to compare the correlations between the variables among the different studies, we first obtained the Fisher Z weighted by the sample size. Then, we transformed the Fisher Z to the correlations by Psychometrica [45]. To evaluate the mediational model, we used structural equation modeling with LISREL 9 and Maximum Likelihood (ML) as the estimation method.

For the mediation analyses, the measures from the cross-sectional studies and only the first measure from each one of the two longitudinal studies were included in this review.

## 3. Results

Following the process of searching for and selecting documents, 11 studies were included, all published between 2010 and 2022 (Table 1). Most of the selected studies (n = 81%) were cross-sectional, while two (n = 19%) were longitudinal.

The selected studies included a total of 25,638 participants aged between 10 and 35 years, with the mean age of participants in the included samples ranging from 13.6 to 23.6. Regarding gender, 10 studies provided detailed percentages; overall, the percentage of female participants was higher than that of male ones (55%). The sample size ranged from 74 to 11,393 participants.

Regarding the origins of the samples, four studies were from the United Kingdom, three from China, two from the United States, one from South Korea, one from Iran, and one from Spain. All the included studies employed non-probabilistic sampling in the selection of participants. Out of the total 25,638 participants, the samples were primarily composed of high school students (78.22%), young adults with unspecified occupations (17.23%), university students (3.37%), students with unspecified educational levels (0.87%), and, in one study, a community sample of adolescents (0.32%).

Regarding the instruments used to assess the variables of defeat, entrapment, depressive symptomatology, and suicidal ideation, it is notable that all studies employed self-report measures, with an average of 21 items per scale and response options of the Likert type. Tarsafi et al. [43] assessed the presence of suicidal ideation using a single ad hoc item: ‘I have had thoughts of suicide in the past’. Similarly, Ren et al. [46] used the item ‘Have you had suicidal ideation in the last 12 months?’. A 6-point scale was provided for the responses, from 1 = never through to 2 = once, 3 = twice, 4 = three times, 5 = four times, and 6 = five times or more.

To assess the reliability of the instruments used in the student sample for measuring defeat and entrapment, 10 studies (90%) employed the original Defeat and Entrapment Scale [12]. The Defeat Scale showed a Cronbach’s alpha of 0.94, indicating strong internal consistency. The Internal Entrapment subscale reported Cronbach’s alpha values ranging from 0.93 to 0.95, while the External Entrapment subscale ranged from 0.88 to 0.96, both reflecting high internal consistency. Pollak et al. [47] used the Short Defeat and Entrapment Scale (SDES) [48], a short version of the original scale, which showed excellent internal consistency in its original validation, with Cronbach’s alpha values ranging from 0.88 to 0.94. However, the internal consistency for the sample in their study was not reported.

For the assessment of depressive symptomatology, four studies (36%) used the Beck scales: the Beck Depression Inventory (BDI) [49] and the Beck Depression Inventory-II (BDI-II) [50], both showing a good internal consistency. The authors reported a mean alpha coefficient of 0.87 for the BDI and 0.92 to 0.95 for the BDI-II [50] in their samples. Three studies (27%) used the Short Depression Anxiety Stress Scale (DASS-D) [51] and the 21-item Depression, Anxiety, and Stress Scale (DASS-21) [52], which are two versions of the original Depression, Anxiety, and Stress Scale (DASS) [53], with a Cronbach’s alpha of 0.86 to 0.93 for the depression subscale in their samples. Lastly, Park et al. [54] used the Depression Scale (CES-D) [55], showing a Cronbach’s alpha of 0.89 in its sample.

The reliability of the instruments used for assessing suicidal ideation was generally good, although the use of different assessment tools led to some variability. The Beck Scale for Suicidal Ideation (BSSI) [56] had a Cronbach’s alpha of 0.86 in the study by Wetherall et al. [30], the Plutchik Suicide Risk Scale (PSRS) [57] had an alpha of 0.78 in the study by Ordóñez-Carrasco et al. [58], and the alpha for the Suicidal Behaviors Questionnaire Revised (SBQ-R) [59] ranged from 0.83 to 0.93 depending on the study. The Suicidal Ideation Questionnaire (SIQ) [60] had an alpha of 0.97, while the Suicidal Ideation Attributes Scale (SIDAS) [61] and the Scale for Suicide Ideation (SSI) [62] both had alphas of 0.90.

A detailed outline of the specific instruments used in each study is provided in Table 1.

**Table 1 behavsci-14-01145-t001:** Summary of the articles included in the systematic review.

Reference	Year	Country	Study Sample (N, Average Age, SD, Sex)	Sample Type	Study Variables	Instruments Used	Results
Bradford et al. [63]	2021	UK	N = 259; Mean Age = 21.15, SD = 2.11; Sex = 34.4% Men	Students and non-students	Depression symptoms, defeat, entrapment, and suicidal ideation	Defeat and Entrapment Scale, DASS-21, SIDAS	r DS/E = 0.69; r DS/D = 0.67; r DS/I = 0.53; r E/D = 0.67; r E/I = 0.55; r D/I = 0.54
Li et al. [64]	2020	China	N = 1239; Mean Age = 14.07, SD = 1.54; Sex = 45.2% Men	Students	Depression symptoms, defeat, entrapment, and suicidal ideation	Entrapment Scale, DASS-21	r DS/E = 0.54; r DS/D = 0.38; r DS/I = 0.34; r E/D = 0.31; r E/I = 0.39; r D/I = 0.19
Ordóñez-Carrasco et al. [58]	2021	Spain	N = 644; Mean Age = 25.91, SD = 5.14; Sex = 48.8% Men	General population	Defeat and entrapment, suicidal ideation, and depression	Defeat and Entrapment Scale, PSRS, BDI-II	r E/I = 0.52
Park et al. [54]	2010	Korea	N = 11,393; Mean Age = 14.62, SD = 1.50; Sex = 56.5% Men	Students	Depression symptoms, entrapment, and suicidal ideation	SSI, CES-D, Entrapment Scale	r DS/E = 0.71; r DS/I = 0.55; r E/I = 0.59
Pollak et al. [47]	2021	USA	N = 74, Mean age = 16.27, Sex = Men–Women, SD = 2.21	Non-clinical sample, general population	Defeat and entrapment, suicidal ideation and depression	SDES, SIQ, QIDS-SR	r DS/E = 0.77; r DS/D = 0.77; r DS/I = 0.60; r E/I = 0.71; r D/I = 0.71
Ren et al. [46]	2018	China	N = 1074; Mean Age = 13.83, SD = 1.53; Sex = 45.8% Men	Students	Depression symptoms, entrapment, and suicidal ideation	Entrapment Scale, DASS-D	r DS/E = 0.6; r DS/I = 0.44; r E/I = 0.38
Tarsafi et al. [43]	2015	USA/Iran	N = 194 (USA); Mean Age = 21.6, SD = 5.2 years; Sex = 23.20% Men N = 146 (Iran); Mean Age = 21, SD = 1.7 years; Sex = 28.08% Men	Students	Depression symptoms, defeat, entrapment, and suicidal ideation	Defeat and Entrapment Scale and BDI	N USA: r DS/E = 0.77; r DS/D = 0.7; r DS/I = 0.54; r E/D = 0.81; r E/I = 0.46; r D/I = 0.45 N Iran: r DS/E = 0.72; r DS/D = 0.72; r DS/I = 0.41; r E/D = 0.73; r E/I = 0.37; r D/I = 0.34
Taylor et al. [65]	2010	UK	N = 93; Mean Age = 23.45, SD = 7.06; Sex = 18.28% Men	University students	Defeat, entrapment, and suicidal ideation	Defeat and Entrapment Scale, SBQ-R	r E/D = 0.73; r E/I = 0.45; r D/I = 0.49
Taylor et al. [25]	2011	UK	N = 79; Mean Age = 19.61, SD = 4.45; Sex = 16.46% Men	University students	Depression symptoms, defeat, entrapment, and suicidal ideation	SBQ-R, BDI-II, Defeat and Entrapment Scale	r DS/E = 0.73; r DS/D = 0.73; r DS/I = 0.62; r E/D = 0.79; r E/I = 0.6; r D/I = 0.6
Yang et al. [66]	2022	China	N = 4515; Mean Age = 15.24, SD = 1.66; Sex = 50.23% Men	High school students	Defeat and suicidal ideation	DSI-SS, Defeat Scale	r D/I = 0.43
Wetherall et al. [30]	2022	Scotland	N = 3508	General population	Defeat and entrapment, suicidal ideation, and depression symptoms	BSSI, BDI-II, Defeat and Entrapment Scale	r DS/D = 0.84; r DS/I = 0.58; r D/I = 0.53

Note. SD: standard deviation; BDI: Beck Depression Inventory [49]; BDI-II: Beck Depression Inventory-II [50]; BSSI: Beck Scale for Suicidal Ideation [56]; CES-D: Depression Scale [55]; DASS-D: Short Depression Anxiety Stress Scale [51]; DASS-21: 21-item Depression, Anxiety, and Stress Scales [52,53]; D: Defeat; DS: Depressive Symptom; DSI-SS: Depressive Symptom Index-Suicidality Subscale [67]; E: Entrapment; I: Suicidal Ideation; PSRS: Plutchik Suicide Risk Scale [57]; QIDS-SR: Quick Inventory of Depressive Symptomatology-Self Report [68]; SBQ-R: Suicidal Behaviors Questionnaire Revised [59]; SIQ: Suicidal Ideation Questionnaire [60]; SDES: Short Defeat and Entrapment Scale [48]; SIDAS: Suicidal Ideation Attributes Scale [61]; SSI: Scale for Suicide Ideation [62].

An analysis of the (normalized) correlations between the variables across the selected studies is presented in Table 2. Although the effect size of these correlations is small, they are statistically significant. Beyond the quantitative value of the relationship, these results suggest the qualitative importance of these associations.

In accordance with the second objective, the mediating effect of depressive symptomatology in the relationship between defeat, entrapment, and suicidal ideation was explored (see Figure 2).

The tested structural equation model showed an excellent fit: χ^2^ = 0.42 (*p* = 0.93; *df* = 3), GFI = 0.99, CFI = 1.00, ECVI = 0.006, RMSEA = 0.00 (90% CI [0.0; 0.008]), and SRMR = 0.005. This model is consistent with the idea that entrapment and defeat act as predictors of depressive symptomatology, which in turn contributes to the emergence of suicidal ideation in adolescents and young people. Suicidal ideation would thus be the result of both the direct effects of entrapment and depressive symptomatology and the indirect effects of entrapment (c = 0.015; CR = 4.79) and defeat (c = 0.016; CR = 4.59) through depressive symptomatology. This indicates a total mediating effect for the defeat variable and a partial mediating effect for entrapment.

## 4. Methodological Quality and Risk of Bias

### 4.1. Methodological Quality of the Analyzed Studies

For the analysis of methodological quality, the MINORS tool [69] was employed as the review included non-randomized and non-comparative studies. For all 11 of the studies included, the overall quality of the trials was considered, taking into account aspects such as randomization, participant dropout/attrition, selective reporting of results, and other potential sources of bias (e.g., sources of funding). The results of the complete checklist are presented in Table 3.

### 4.2. Methodological Quality of the Review

The systematic review adhered to PRISMA-P criteria [39,40] and followed the bias control guidelines recommended by AMSTAR 2 [42]. The methodological quality of the systematic review was assessed using the AMSTAR 2 tool [42], which evaluates confidence levels across 16 criteria. The overall confidence level was deemed to be moderate, with only a few limitations identified, such as the lack of a more comprehensive literature search method.

## 5. Discussion

This study conducted a systematic review of the literature investigating the association between the variables of defeat, entrapment, and depressive symptoms in relation to suicidal ideation in adolescents and young adults utilizing formal search strategies. The use of a comprehensive search strategy that is easily replicable in future research facilitated the thorough exploration of empirical findings on the role of defeat and entrapment in suicidal behavior within the child and adolescent population.

The most noteworthy result from the selected studies is the consistent association between the key variables highlighted in the Integrated Motivational–Volitional (IMV) model [10]. Specifically, the model underscores the role of defeat as a variable implicated in the formation of entrapment, which is, in turn, closely linked to the development of suicidal ideation in the general population [58].

This evidence supports the theoretical framework of the IMV model [10] and aligns with other studies that have tested the relationships among the different factors within the model [70]. It also supports models that focus on identifying psychological states preceding a suicide attempt and factors that elevate the risk of suicidal behavior, such as those proposed by Galynker et al. [71]. The contribution of defeat and entrapment as variables has also been confirmed in the review by Taylor et al. [13], which indicated that these factors were associated with an increased risk of suicide in both clinical and community populations.

Specifically, when analyzing the interplay between the three variables, several studies included in this review demonstrate that defeat and entrapment are robustly associated with measures of depressive symptoms and suicidal ideation, regardless of the country under study [43,47]. This suggests the universal presence of defeat and entrapment, independent of cultural context or age, as significant factors in explaining potential mechanisms behind suicidal behavior. These variables may also serve as relevant indicators of suicide risk.

Furthermore, as highlighted by various studies, both perceptions of defeat and entrapment are strongly associated with the risk of self-harm [72] and suicidal thoughts and behaviors [73].

This review is grounded in O’Connor’s [10] Integrated Motivational–Volitional (IMV) model, which explores the variables of entrapment and defeat and their association with suicidal ideation in child and adolescent populations. The IMV model proposes that suicidal behavior results from dynamic and reciprocal interactions among various factors, some of which may act as mediators in the relationship between the core variables implicated in suicidal behavior. Studying the interaction between these variables can suggest mechanisms that can explain the dynamics of suicide in adolescents and young adults. Depressive symptomatology—particularly those components related to future expectations and hopelessness—has been interpreted as being directly linked to suicidal ideation, as argued by Beck et al. [20] from a cognitive perspective. It also acts as a potential mediating variable between the motivational and volitional phases of suicide risk. According to the IMV model [11], this depressive symptomatology may facilitate the transition from suicidal ideation to action. The findings of this review and the proposed model support these concepts in the child and adolescent population, contributing to a better understanding of the role of depressive symptoms in suicidal dynamics. Consequently, this contributes to early detection as well as the improvement of prevention and intervention strategies in clinical, educational, and other settings [33,74].

These results, when considered globally, provide evidence supporting both the IMV model [11], which posits a central role for perceived defeat and entrapment as key variables in the motivational phase of suicidal ideation development, and Beck et al.’s [20] model, where depressive symptomatology plays a crucial role as a precursor to suicidal thoughts. Together, these findings reinforce the understanding that both perceived defeat and depressive symptoms are pivotal factors in the trajectory toward suicidal ideation, emphasizing the need to address them in prevention and intervention efforts.

As for the methodological quality of the studies analyzed, the overall level of confidence is moderate since only a few critical weaknesses have been detected. An accurate summary is provided of the available state-of-the-art studies. The systematic review conducted followed the PRISMA-P criteria [39,40] and the guidelines for bias control indicated by AMSTAR 2 [42].

## 6. Limitations and Future Directions

Despite the strengths of this study, it is necessary to highlight some of its limitations. One of them, primarily methodological in nature, stems from the undefined number of studies that may have been excluded from the database search; specifically, in ProQuest, the search was limited to abstracts, whereas this restriction was not applied across the other databases, WoS and PubMed. A second limitation arises from the inclusion and exclusion criteria themselves, as studies that did not specify certain variables, such as the ages of the participants, were discarded. Additionally, it is important to emphasize that the published literature in this area remains scarce due to the ethical and legal requirements involved in studying suicidal behavior in minors. Furthermore, the limited availability of longitudinal studies constrains the capacity to establish causal inferences.

In light of all this and given the shortcomings of the review realized, the results suggest the need for further research in this field, including both cross-sectional studies with the general population and longitudinal studies in diverse cultural contexts as well as studies at different stages of adolescent development, that allow us to carry out meta-analysis using structural equation modeling to obtain more realistic empirical evidences about the tentative model proposed. This would help, firstly to properly stablish causal relationship, and consequently shed light on the processes involved in the transition from ideation to attempt and lethal suicidal action, clarifying the contributions of defeat and entrapment in this process.Despite the methodological limitations and the scarcity of longitudinal studies that examine the transition from suicidal ideation to attempt, the present work provides a solid foundation for future research and preventive actions. It is worth noting that research on suicide prevention has evolved from attempting to establish causal relationships to focusing on plausible explanatory models that enhance prevention efforts [9]. Thus, the consistent relationships found between defeat, entrapment, depressive symptoms, and suicidal ideation underscore the importance of addressing these variables in the early detection of and intervention in this issue, whose prevalence has been increasing in recent years [1,2]. Ultimately, addressing these factors comprehensively may significantly contribute to reducing the risk of suicide in the child and adolescent population, thus improving the effectiveness of prevention and treatment strategies in clinical and educational settings.

## Figures and Tables

**Figure 1 behavsci-14-01145-f001:**
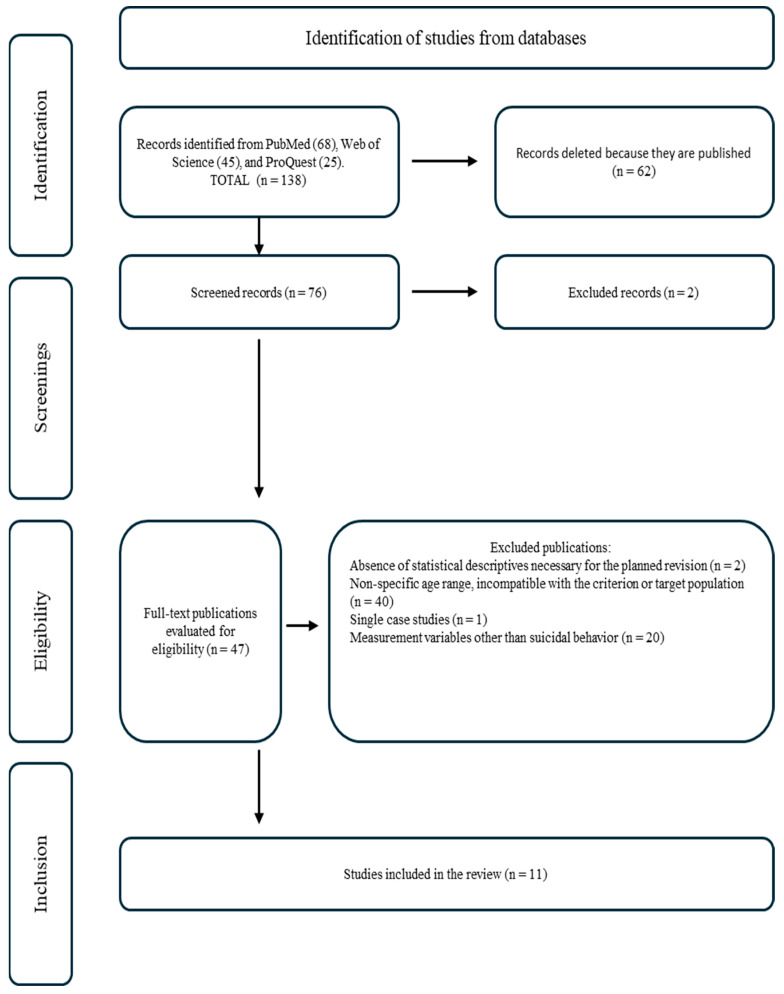
Flow diagram according to the PRISMA Model for Systematic Review.

**Figure 2 behavsci-14-01145-f002:**
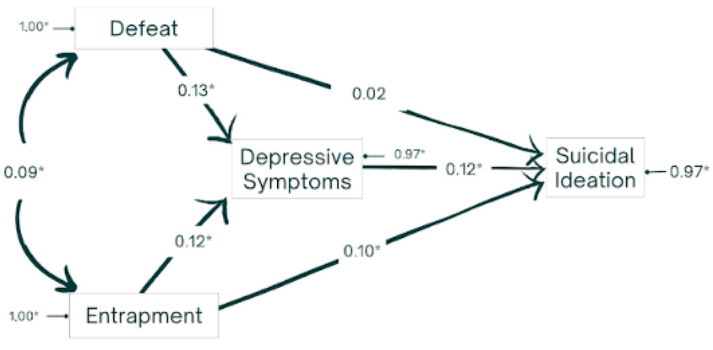
Standardized solution of the tentative model. Note. * *p* < 0.05.

**Table 2 behavsci-14-01145-t002:** Normalized correlations calculated from the selected studies.

	1	2	3	4
Ideation (1)	-			
Depressive symptomatology (2)	0.14	-		
Defeat (3)	0.05	0.14	-	
Entrapment (4)	0.12	0.13	0.09	-

**Table 3 behavsci-14-01145-t003:** The revised and validated version of MINORS.

Methodological Items for Non-Randomized Studies	Score †
1. A clearly stated aim: The question addressed should be precise and relevant in light of the available literature.	2
2. Inclusion of consecutive patients: All patients potentially fit for inclusion (satisfying the criteria for inclusion) should have been included in the study during the study period (there should be no exclusion or details about reasons for exclusion).	2
3. Prospective collection of data: The data should have been collected according to a protocol established before the beginning of the study.	2
4. Endpoints appropriate to the aim of the study: There should be an unambiguous explanation of the criteria used to evaluate the main outcome, which should be in accordance with the question addressed by the study. Also, the endpoints should be assessed on an intention-to-treat basis.	2
5. Unbiased assessment of the study endpoint: There should be blind evaluation of objective endpoints and double-blind evaluation of subjective endpoints. Otherwise, the reasons for not blinding should be stated.	not applicable
6. Follow-up period appropriate to the aim of the study: The follow-up should be sufficiently long to allow for assessment of the main endpoint and possible adverse events.	1
7. Loss to follow-up less than 5%: All patients should be included in the follow-up. Otherwise, the proportion lost to follow-up should not exceed the proportion experiencing the major endpoint.	not applicable
8. Prospective calculation of the study size: There should be information on the size of detectable difference in interest with a calculation of a 95% confidence interval, according to the expected incidence of the outcome event, and information about the level for statistical significance and estimates of power when comparing the outcomes.	0

† The items are scored 0 (not reported), 1 (reported but inadequate), or 2 (reported and adequate). The global ideal score is 16 for non-comparative studies and 24 for comparative studies. The total score obtained was 9 of out 16 points.

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
