# Peer review of "Suicidal Ideation in Adolescents and Young Adults: The Role of Defeat, Entrapment, and Depressive Symptoms—From a Systematic Review to a Tentative Theoretical Model"

_behavsci, 2024, doi:10.3390/bs14121145_

Round 1
Reviewer 1 Report
Comments and Suggestions for Authors
1 Summary of the research and overall impression
This review explores available quantitative studies in clinical and non-clinical populations, age between 10 and 35 years. It tests the role of depression in suicidal behavior, in relation to the IMV model. It finds an association between entrapment and defeat, leading to (more) depressive symptoms, subsequently inducing higher suicidal ideations.
The study is well-described, carefully referenced. It has some limitations that were not reported, like leaving out longitudinal data, and tested only one tentative model, perhaps excluding a clinically meaningful alternative. Improvement on this points is recommended.
2 Discussion of specific areas of improvement
Major points:
A. The discussion of limitations leaves out an important limitation: no longitudinal data have been included. 2 studies reported followup assessments, but these were left out. What do these outcomes show, and does this align with the current findings? But more importantly, the causal relationships that are suggested, could not be based on only correlational data. This has to be mentioned anywhere in the discussion. Also in the quality assessment, this needs attention: no study can have any points on the MINORS item 6.
B. Figure 2: Only one tentative model has been tested. Clinically, a model with depressive symptoms inducing entrapment and defeat would make as much sense. Such a model could be tested too, using the same data, as far as I can see. I suggest adding this model as alternative model: depress symptoms > defeat; depress symptoms > entrapment; entrapment > SI; defeat > SI
C. In the recommendations, a recommendation for longitudinal research is lacking, but needed for higher quality conclusions on the theoretical conceptualizations at hand.
Minor points:
a. Line 134: Results were limited to search terms in abstracts. This limits the number of outcomes, and makes the work lighter, but also excludes relevant studies. In line 338 this needs some mention.
b. Line 188: The high prevalence of high school students in the total sample perhaps induces a selection bias?
c. Lines 195-198: It is not clear to the reader what study used which question.
d. Results section: Are all reported alpha’s from the included studies or from original reports like Becks BDI studies? Perhaps in line 198 or around there, some mentioning of the origin of all alpha’s would be helpful to clarify this.
e. Line 116/117: It is not clear why and how AMSTAR was used. The MINORS tool (line 261) seems to be used instead. AMSTAR would be of no use to evaluate crosssectional studies, it is meant to evaluate reviews. Additionally, it has been updated to AMSTAR 2: AMSTAR 2: a critical appraisal tool for systematic reviews that include randomised or non-randomised studies of healthcare interventions, or both | The BMJ
f. Table 2: Correlations in table 2 are relatively low, although statistically significant. The relative weight of these correlations has not been worded anywhere, as far as I can see. Perhaps this can be added.
Author Response
Dear reviewer, enclosed are the responses to each of your valuable comments. We would like to thank you for your review and also appreciate all your suggestions to improve our first version of the manuscript.
Thank you very much
Sincerely

Reviewer 2 Report
Comments and Suggestions for Authors
The abstract and literature review are exemplary. Detailed descriptions of the IMV model were provided and the literature was appropriate, concise and comprehensive. One strength of your paper is that you utilized a global review of scientific studies. The methods used to accomplish this search are sound and exhaustive. Your methodology is sound, my one concern was that there were so many exclusion criteria which you do later mention as a limitation. That being said, even with a sample size of 11 studies it totaled 25,638 participants. Data analysis and presentation of the results indicate that the chosen variables are associated and bolster the IMV model. The discussion section is consistent with the evidence presented from detailed analytical techniques. The overarching benefit of this review is that it significantly contributes to the literature on suicide and clearly indicates that the IMV theoretical model is worthy of further study to determine how programs of intervention, in particular, could address the three phases: premotivational, motivational, and volitional, and determine where preventive interventions could best be utilized as in “what stage”. Further research is needed to investigate how the variables (defeat, entrapment, depressive symptoms, and suicidal ideation) lead to each phase of the IMV model and at which phase would prevention be most appropriate.
Author Response

(The authors gave the same response as above.)
